# *Legionella pneumophila*—Epidemiology and Characterization of Clinical Isolates, Slovenia, 2006–2020

**DOI:** 10.3390/diagnostics11071201

**Published:** 2021-07-02

**Authors:** Darja Keše, Aljoša Obreza, Tereza Rojko, Tjaša Cerar Kišek

**Affiliations:** 1Faculty of Medicine, Institute of Microbiology and Immunology, University of Ljubljana, Zaloška 4, 1000 Ljubljana, Slovenia; aljosa.obreza@mf.uni-lj.si (A.O.); Tjasa.Cerar@mf.uni-lj.si (T.C.K.); 2Department of Infectious Diseases, University Medical Centre Ljubljana, Japljeva 2, 1000 Ljubljana, Slovenia; tereza.rojko@kclj.si

**Keywords:** *Legionnaires’* disease, diagnosis, *Legionella pneumophila* typing, cultivation, epidemiology

## Abstract

*Legionella pneumophila* is the causative agent of severe Legionnaires’ disease (LD). Although an increasing number of LD cases have been observed, published data from Slovenia are very limited and data on molecular epidemiology are even scarcer. The present retrospective study (2006–2020) reports the results of the microbiological diagnosis of LD, as well as the epidemiology and characterization of the *Legionella* clinical isolates. We tested urine samples from 15,540 patients with pneumonia symptoms for *L. pneumophila* infection by urine antigen test, of which 717 (4.6%) tested positive. Isolation of *L. pneumophila* was successfully performed from 88 clinical specimens, with 82 (93.2%) being identified as *L. pneumophila* sg 1 and six (6.8%) as *L. pneumophila* sg 2–14. Sequence-based typing (SBT) identified 33 different sequence types (STs), the most frequent being ST1 and ST23. Sequence type 1 mainly comprised isolates belonging to the Philadelphia subgroup, and ST23 mostly to Allentown/France. The standard SBT scheme, as well as Dresden phenotyping for *L. pneumophila,* presented a high diversity among isolates.

## 1. Introduction

*Legionella pneumophila* is a Gram-negative rod-shaped bacteria that is ubiquitous in the natural aquatic environment. They are facultative intracellular parasites of free living protozoa, mostly amoeba, and they can also occasionally infect humans. Infection with *Legionella* occurs following inhalation of contaminated aerosols produced by many man-made water systems such as cooling towers, air conditioners, showers, faucets, baths and spas [1]. In hospitals, the important route of infection is microaspiration of drinking water contaminated with *Legionella* or by direct incorporation into the lung during respiratory tract manipulation [2,3]. After *Legionella* enters the pulmonary alveoli, it is phagocytosed by macrophages, where exponential replication occurs [4]. Such an infection leads to the development of Legionnaires’ disease (LD), a severe, life-threatening pneumonia, or to a flu-like illness called Pontiac fever. LD accounts for approximately 2–15% of all cases of community-acquired pneumonia (CAP) requiring hospitalization in Europe and North America [5]. It is known that *L. pneumophila* can cause sporadic and epidemic CAP, as well as hospital-acquired pneumonia (HAP), in both healthy and immunocompromised hosts [3,6]. In addition, LD also represents a significant cause of travel associated respiratory tract infections [1]. A confirmed laboratory diagnosis of LD is based on the detection of *L. pneumophila* antigen in the urine, the isolation of *Legionella* spp. from lower respiratory secretions or any normally sterile site, or the demonstration of a significant rise in specific antibody level to *L. pneumophila* sg 1 in paired patient sera samples [7].

LD is notifiable in European Union Member countries, as well as in Iceland, Norway, Switzerland, and the U.K. [8,9]. The surveillance of LD at the European level began in 1986 and, since 2010, has been carried out by the European Legionnaires’ Disease Surveillance Network (ELDSNet), the successor of EWGLI (European Working Group for *Legionella* Infections), and coordinated by the European Centre for Disease Prevention and Control (ECDC) [10]. 

In Slovenia, notification of confirmed *Legionella* infections to the National Institute of Public Health (NIPH) has been mandatory since 1990; additionally, notification of LD cases to EWGLI–ELDSNet has been carried out since 1999. In the last 15 years, an increasing number of LD cases has been observed in Slovenia, mostly diagnosed by the detection of *L. pneumophila* antigen in the urine. According to ELDSNet Annual Meeting Reports [11] the notifications rates in Slovenia are amongst the highest in Europe. This is, in all probability, also a reflection of the growing awareness resulting in diagnostic testing performed in almost all adult patients with pneumonia treated in hospital and in regular reporting. Nevertheless, published LD data in Slovenia are very limited and data on molecular epidemiology are even scarcer. 

The aim of the present retrospective study was to present the results of an epidemiological investigation and microbiological diagnosis of LD in our patients and to characterize the clinical isolates isolated between 2006 and 2020 using phenotyping and genotyping methods.

## 2. Materials and Methods

### 2.1. Sample Collection and Routine Legionella Testing

From January 2006 to December 2020, urine samples from 15,540 patients with pneumonia symptoms were obtained for testing for *L. pneumophila* infection. The patients were 18–87 years of age (median age = 62). They were treated mainly in the University Medical Center Ljubljana (UMCL), which is the leading medical institution in Slovenia, and in some regional hospitals. The patients were inhabitants of the largest Central Region of Slovenia and three smaller Statistical regions, which, on average, represent 38.79% (803,095/2,070,275) [12] of the whole Slovenian population during the designated period of time. Depending on the severity of symptoms, patients were treated as inpatients or outpatients. Urine samples were collected from a total of 9200 males and 6340 females. Additionally, lower respiratory tract samples such as sputum, bronchoalveolar lavage, or tracheal aspirate were available from 3038/15,540 (19.5%) and were tested by real-time PCR and culture. For the presented study, all human samples were anonymized and the data pertaining to the patients’ gender, age, and region of origin were linked only to randomized numerical codes. 

### 2.2. Urine Antigen Test (UAT)

Urine samples were analyzed by the *Legionella* Urinary Antigen EIA (UAT) test. Biotest *Legionella* urine antigen EIA (Biotest AG, Dreieich, Germany) was used until 2011 for detecting the antigens of all *L. pneumophila* serogroups. This was later replaced by Binax kits (Alere Scarborough, Inc., Scarborough, ME, USA), which are specific for *L. pneumophila* sg 1 antigen. Both tests were performed according to the manufacturer’s instructions.

### 2.3. Polymerase Chain Reaction (PCR)

DNA was extracted from 200–1000 µL of lower respiratory samples using the automated MagNA Pure Compact System (Roche Diagnostics, Mannheim, Germany) and the MagNA Pure Compact Nucleic Acid Isolation Kit I (Roche Diagnostics, Mannheim, Germany) with Bacteria Lysis Buffer (Roche Diagnostics, Mannheim, Germany) and proteinase K (Roche Diagnostics, Mannheim, Germany) pretreatment. DNA was eluted to a final volume of 100 µL. Real-time PCR targeting the *mip* gene of *L. pneumophila* (Argene-BioMerieux diagnostics, Marcy l’Etoile, France) was performed on a LightCycler 2.0 (Roche Diagnostics, Mannheim, Germany) platform according to the manufacturer´s instructions. PCR-positive samples were further analyzed by interlaboratory-validated *L. pneumophila* triplex *qPCR* assay to discriminate between *L. pneumophila* sg 1 (*wzm* target gene) and *L. pneumophila* (*mip* gene) on the Rotor-Gene Q (Qiagen GmbH, Hilden, Germany) platform as previously described [13]. In both PCR tests, internal controls were included.

### 2.4. Legionella spp. Cultivation

*Legionella* cultivation was subsequently performed only on the remaining respiratory samples from PCR-positive patients. Sputum samples were liquefied with Sputasol (OXOID, Basingstoke, UK) in a 1:1 ratio. After centrifugation at 1000× *g* for 15 min, the supernatant was discarded. The resulting pellet was resuspended in 1 mL of sterile H_2_O and heat-treated at 50 °C for 30 min. Buffered charcoal–yeast extract agar (BCYE) and *Legionella*-selective (BMPA) agar plates with antibiotics (OXOID, Basingstoke, UK) were inoculated using the standard technique [14]. Plates were incubated aerobically in a humid atmosphere at 35–37 °C for up to 10 days with regular macroscopic and microscopic inspections for specific *Legionella* colonies every two days. Presumptive *Legionella* colonies that appeared after the third day with a round, smooth shape and a bluish-gray, or whitish appearance were identified using a Microflex LT (Bruker Daltonics, Billerica, MA, USA) MALDI-TOF MS system according to the manufacturer´s instructions. Data acquisition and result interpretation were automatically performed by the MALDI Biotyper software (Bruker Daltonics, Billerica, MA, USA). All *Legionella* isolates were identified by indirect immunofluorescence test (IFA) using specific monoclonal antibodies and then stored on −80 °C.

### 2.5. Phenotypic and Genotypic Analysis

Identification to *L. pneumophila* sg 2–14 was first performed using the MonoFluo *Legionella pneumophila* IFA Test (Bio-Rad Laboratories, Marnes-la-Coquette, France), while *L. pneumophila* sg 1 was determined by DFA reagent (Pro-Lab, Richmond Hill, ON, Canada) following the manufacturer’s instructions.

Subsequently, for research purposes, the Dresden panel of monoclonal antibodies (MAb) was used by IFA to further characterize the exact serogroup or the phenotypic subgroup of *L. pneumophila* sg 1 [15]. The slides were inspected under an Eclipse E400 (Nikon, Tokyo, Japan) fluorescence microscope.

All *Legionella* isolates were typed using the sequence-based typing (SBT) according to the EWGLI standard protocol, which is based on amplification and sequencing of seven selected gene loci (*flaA*, *pilE*, *asd*, *mip*, *mompS*, *proA,* and *neuA)* that define the allelic profile and assignment of sequence types (STs) [16,17]. The nested SBT protocol was used for PCR-positive clinical samples, from which the *Legionella* isolate was not obtained [18]. Sequences were obtained by Sanger sequencing using the BigDye Terminator v3.1 Cycle Sequencing Kit in the ABI 3500 Genetic Analyzer (Applied Biosystems, Foster City, CA, USA) and submitted to the EWGLI *Legionella* SBT database in Public Health England.

A minimum spanning tree (MST) based on STs was constructed using Bionumerics software (Bionumerics ver. 8.0; Applied Maths NV—a bioMérieux company, Sint-Martens-Latem, Belgium).

### 2.6. Serology

Serum samples from 6175 of the 15,540 patients (39.7%) were available, and immune response to *Legionella pneumophila* infection was investigated by IFA (R-Biopharm AG, Darmstadt, Germany). A fourfold seroconversion to ≥1:128 in paired serum samples and a single titer ≥1:512 were diagnostic.

## 3. Results

### 3.1. Microbiological Diagnostics of Legionella pneumophila Infections

During the study period, a progressive increase in the use of urine antigen tests for the diagnosis of LD was noted; on the contrary, the number of serology tests declined. PCR testing remained almost constant, although low, through the years (Figure 1). Namely, in the year 2006, only 306 patients were tested by UAT, 66 by PCR, and 706 patients by serology assay IFA; meanwhile, the corresponding values in the year 2019 were 2182 for UAT, 223 for PCR, and 164 for IFA. In 2020, the number of all tests decreased due to the COVID-19 pandemic (Figure 1). Altogether, anti-*Legionella pneumophila* antibody response, as seroconversion or high single antibody titer (≥1:512), was identified in 213 of the 6175 (3.4%) tested sera. *Legionella* EIA UAT was performed for a total of 15,540 patients, of which 717 (4.6%) tested positive; 516 (72.0%) of them were male and 201 (28.0%) female, with a male/female ratio of 2.6 (Figure 2). 

The number of *Legionella* cases progressively increased during the study years, but the positivity rate for each year during the assessed time period was relatively stable, except for the year 2008 (7.7%), when an outbreak of legionellosis was recognized, and the year 2012 (Figure 3). It is noticeable that the number of LD cases started rising in the months of May and June each year, reaching a peak in July and staying high until November (Figure 4). 

Lower respiratory tract samples were available from only 3038 patients, of which 129 (4.2%) were positive by *Legionella pneumophila* PCR: 118 (91.5%) were characterized as *L. pneumophila* sg 1 and 11 (8.5%) as *L. pneumophila* sg 2–14. 

Isolation of *L. pneumophila* was successfully performed in 88 (68.2%) of the 129 PCR-positive cases: 82 (93.2%) were identified as *L. pneumophila* sg 1 and six (6.8%) as *L. pneumophila* sg 2–14. 

### 3.2. Legionella pneumophila Phenotyping and Genotyping Analysis

By using the Dresden panel of monoclonal antibodies, 71 (86.6%) of the *L. pneumophila* sg 1 clinical isolates were shown to react with the more virulent MAb 3/1 associated with the more virulent strains; the remaining 11 (13.4%) isolates did not react with MAb 3/1. The most frequent Dresden subgroups was Allentown/France (36/43.9%), followed by Philadelphia (24/29.3%), Bellingham (9/11.0%), Knoxville (7/8.5%), Benidorm, (4/4.9%) and Olda/Oxford (2/2.4%) (Table 1).

A total of 88 *L. pneumophila* clinical isolates were successfully genotyped using standard SBT; additionally, 10 complete typing profiles were obtained directly from clinical samples by the nested SBT. A total of 33 different STs were identified (Figure 5 and Figure 6). The most frequent were ST1 and ST23 (19.4% each), followed by ST62 (6.1%), ST82 (5.1%), and ST37, ST203, ST728, and ST1852 (4.1% each). All of the other STs were only sporadically identified. Two novel STs, namely, ST2999 and ST2998, were identified using standard SBT and added to the international *Legionella* database. Both of them were *L. pneumophila* sg 2–14. 

The MST produced with the ST data showed six clonal complexes (CCs) (Figure 6). Sequence type 1 was the dominant ST, consisting mainly of isolates belonging to the Philadelphia subgroup. The most prevalent CC1 consisted of 21 isolates assigned to ST23 and ST762 (single-locus variant of ST23). The second most prevalent CC was CC2, consisting of nine isolates assigned to ST1852, ST146, ST435, and ST20, followed by CC3 (ST62 and ST1983) consisting of eight isolates and CC4 (ST728, ST421 and ST2999), consisting of seven isolates. The isolates in clonal complexes 1, 3, 6, and 7 were mostly related to the Allentown/France subgroup. 

## 4. Discussion

This retrospective study presented the first comprehensive reports on microbiological diagnostics for LD in Slovenia and the molecular characterization of *L. pneumophila* strains over the period of 2006–2020. It was shown that the diagnosis of legionellosis is mainly based on UATs, and rarely on PCR, culture, or serology. It was also noted that the use of serology tests declined steeply over time. The applicability of serology as an LD diagnostic and/or confirmatory test is quite limited. Significant seroconversion can only be determined using acute and convalescent sera against *L. pneumophila* sg 1, which is time consuming, while interpretation of the results is subjective and, from single-serum samples is usually controversial [3]. 

The increased usage of UAT is likely a consequence of its advantages for diagnosis; in particular, because of the simplicity of sample collection, the test is highly specific and allows early diagnosis. However, this test has low sensitivity for infections caused by *L. pneumophila* serogroups other than sg 1 and is not able to detect other potentially pathogenic *Legionella* species [3,19]. Nevertheless, the *Legionella* UAT is used in 70% to >90% of diagnostic methods for LD confirmation in Europe and the USA [20,21,22,23]. In Slovenia, the use of PCR testing has slightly increased in recent years; however, it is still low, which is probably due to the fact that *Legionella* patients hardly produce productive coughs. In a large retrospective Belgian (Flemish) multicenter study, the added value of PCR on a respiratory specimen was compared to UATs for the diagnosis of legionellosis: 37.5% (15/40) of infections of *L. pneumophila* were missed when UATs were performed as the sole diagnostic test [24]. Another factor that may also contribute to a more difficult diagnosis of LD by UATs is a higher prevalence of non-*L. pneumophila* in patients with nosocomial infection or in highly immunosuppressed patients [25]. In a recent study, Dagan et al. [5] found that only 53.85% of *Legionella* HAP was diagnosed by UATs compared to 85.7% of the cases in the CAP group. The collection of respiratory samples for the diagnosis of LD by PCR should be considered in those cases where the UAT result is negative but the clinical and epidemiological picture is suggestive of LD. In such cases, PCR testing of lower respiratory tract specimens could improve the diagnosis of patients with *Legionella pneumophila* non-sg 1 infection or with *Legionella* spp.

Our results of *L. pneumophila* UATs of adult patients with pneumonia indicate that the epidemiological pattern of *Legionella* cases is similar to those described in other European countries [8,22,26,27]. Legionellosis was diagnosed in 717 (4.6%) of the 15,540 tested patients, predominantly from older patients (median age = 62), especially men. Most of the cases were sporadic and CAP. The data also show that the number of diagnosed LD cases increased through the study period, which could be due to more extensive testing of pneumonia patients for possible *Legionella* infection, as the proportion of positives samples was more or less constant. As illustrated in the ECDC reports, the annual notification rate in EU/EEA countries also increased continuously over the 2011–2017 period from 1.0 per 100,000 in 2011 to 1.8 in 2017 [11]. A number of factors could possibly contribute to an increase in the number of notified LD cases in Europe, such as climate change with a higher temperature and more rainfall, the growing proportion of older people in the EU/EEA population who are more at risk of LD, and also improvement of the surveillance of LD in some European countries [10].

A higher rate of LD cases was observed in Slovenia during 2008 and 2012. A small outbreak occurred in 2008 in one regional hospital, where four patients were found to have positive *Legionella* UATs and PCR tests. Furthermore, *L. pneumophila* sg 1, subgroup Knoxville, ST1 was isolated from two patients and from water samples. In August 2010, another outbreak was identified in a nursing home, affecting 10 of 234 residents [28]. Patients were diagnosed by positive UATs and seroconversion of antibody titer; *L. pneumophila* sg 1, subgroup Allentown/France, ST23 was isolated from two patients involved in the outbreak, as well as from water samples. In 2012, the highest proportion of positive UAT cases was identified, but no outbreak occurred; in the same year, in Europe, 5856 LD cases were reported from 30 countries. As in previous years, six countries, namely, Italy, France, Spain, Germany, the U.K., and the Netherlands reported the highest number, accounting for 84% of all notified cases. Despite the large number of LD cases in these countries, it is worth noting that Slovenia—with 81 LD cases that year—had the highest notification rate, with 3.74 per 100,000 inhabitants [29]. According to our results and the Slovenian surveillance data, the notification rate slightly decreased in the years 2013 and in 2014; however, it increased again in the following years, reaching a peak in 2018 (7.7 /100,000) [30]. More than half of these cases (57.8%) were diagnosed in our laboratory.

During the study period, only 3038 clinical samples were tested by PCR, and 129 LD cases were identified among them; meanwhile, the isolation of *Legionella* was successful in 88 cases. The standard SBT scheme, as well as Dresden phenotyping for *L. pneumophila,* presented a high diversity among isolates, as previously shown in studies from other countries [31,32,33,34]. ST1 and ST23 were the most common, found in 19 isolates each, belonging to different phenotypes: ST1 mostly to Philadelphia and ST23 mostly to Allentown/France. ST1 is globally spread and is considered the major cause of community- and hospital-acquired LD in many countries [35]. Furthermore, ST1 is by far the most common genotype associated with sporadic cases [33]. Six CCs were identified using the minimum spanning tree, with CC1 being the most common. CC1 includes 19 isolates of ST23 and two of ST762; these results are consistent with previous studies where ST23 has been shown to be among the top five disease-associated STs (1, 23, 37, 47, and 62) responsible for nearly half of all epidemiologically unrelated LD cases worldwide [36]. In our study, we also identified ST37, but only in four cases, and ST62 in six cases; thus far, ST47 has not been isolated in Slovenia. Although, the number of our clinical isolates was relatively low, high genetic diversity was observed.

We are aware of the limitations of our study, including the fact that the study was mainly limited to the Central Region of the country and the results do not necessarily represent the epidemiological situation of the whole country. Unfortunately, since we wanted to present data in the study primarily from a microbiological perspective, we did not collect clinical data for all of the included patients, which is certainly also one of the limitations. Another drawback of this study is the small number of respiratory specimens examined by PCR tests and cultures. Therefore, we do not have any data on *Legionella* non-*pneumophila* infections among our patients. For the future, it would be interesting to see the distribution of *Legionella* strains in our environment and to determine the origin of *Legionella* infections in our patients. 

## Figures and Tables

**Figure 1 diagnostics-11-01201-f001:**
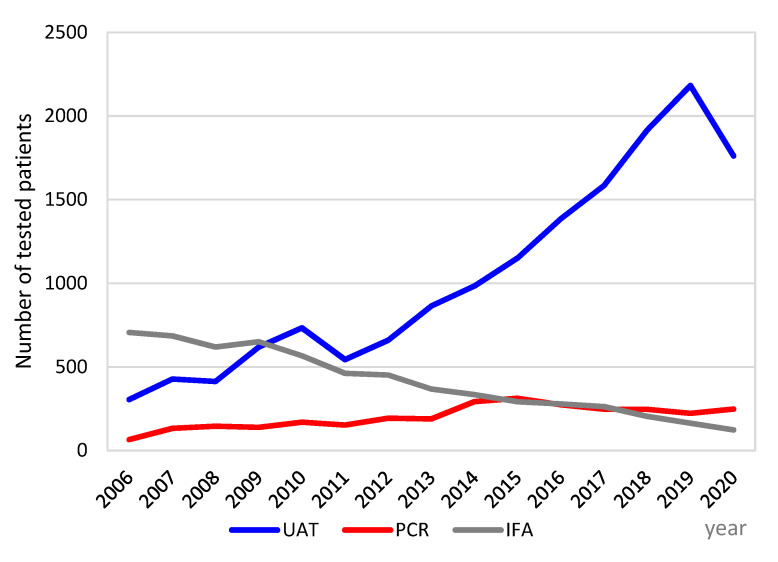
Number of pneumonia patients tested by *Legionella pneumophila* urine antigen test (UAT), *Legionella pneumophila* PCR and *Legionella pneumophila* serology (IFA) per year (Slovenia, 2006–2020). Each patient could be tested with more than one test. UAT, urine antigen test; PCR, polymerase chain reaction; IFA, indirect immunofluorescence assay.

**Figure 2 diagnostics-11-01201-f002:**
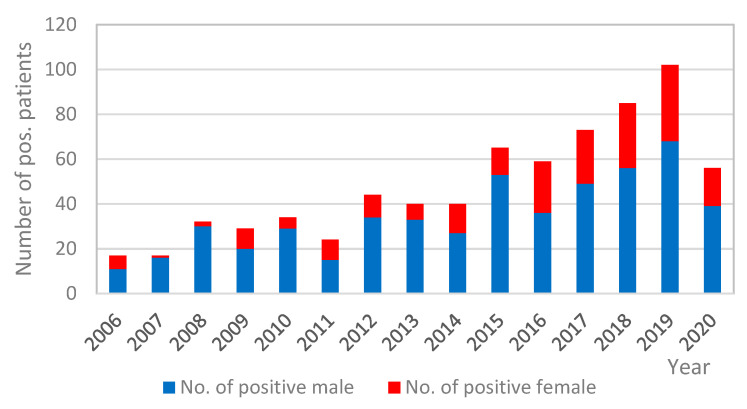
Number of positive male and female patients tested by *Legionella pneumophila* urine antigen test per year (Slovenia, 2006–2020).

**Figure 3 diagnostics-11-01201-f003:**
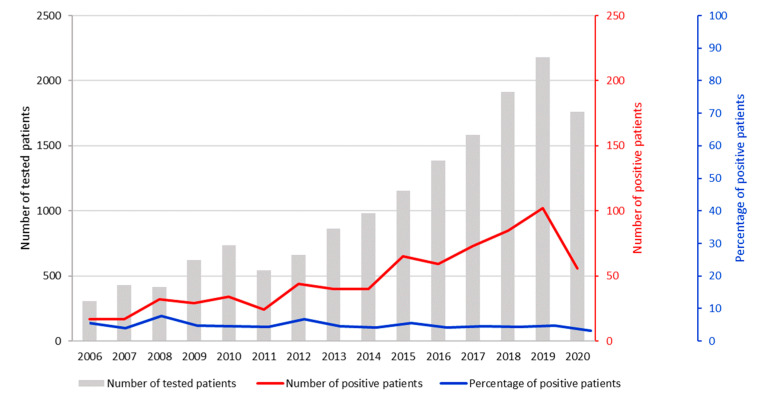
Number of pneumonia patients tested by the *Legionella pneumophila* urine antigen test (2006–2020). The red and the blue lines represent the number and proportion of positive patients per year, respectively.

**Figure 4 diagnostics-11-01201-f004:**
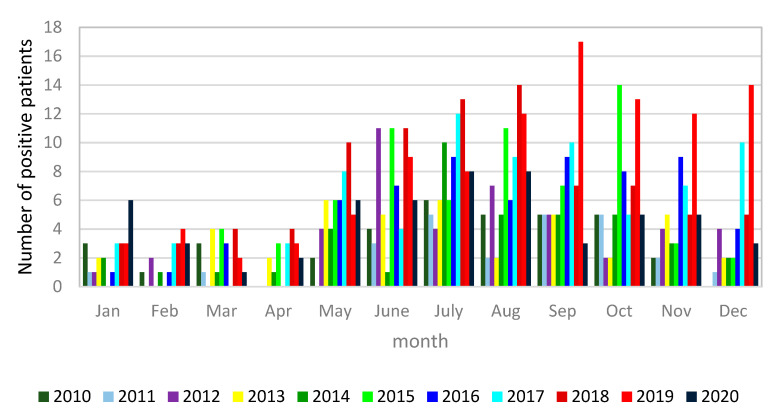
Monthly distribution of *Legionella pneumophila*-positive cases diagnosed by the *Legionella* urine antigen test (2010–2020).

**Figure 5 diagnostics-11-01201-f005:**
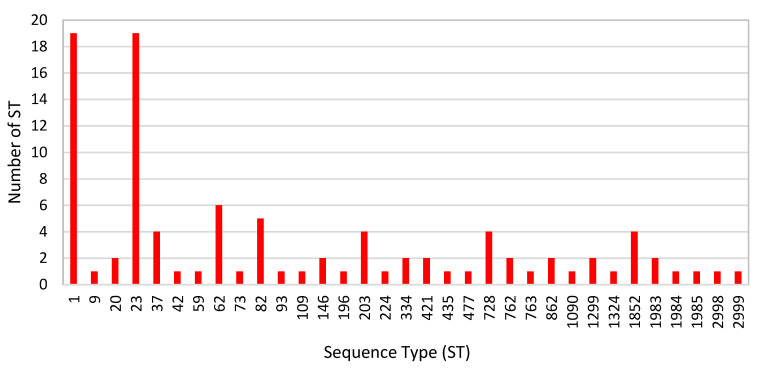
*Legionella pneumophila* sequence types (ST) identified among the clinical samples in the period of 2006–2020 (Slovenia). A total of 98 *Legionella pneumophila* STs are presented.

**Figure 6 diagnostics-11-01201-f006:**
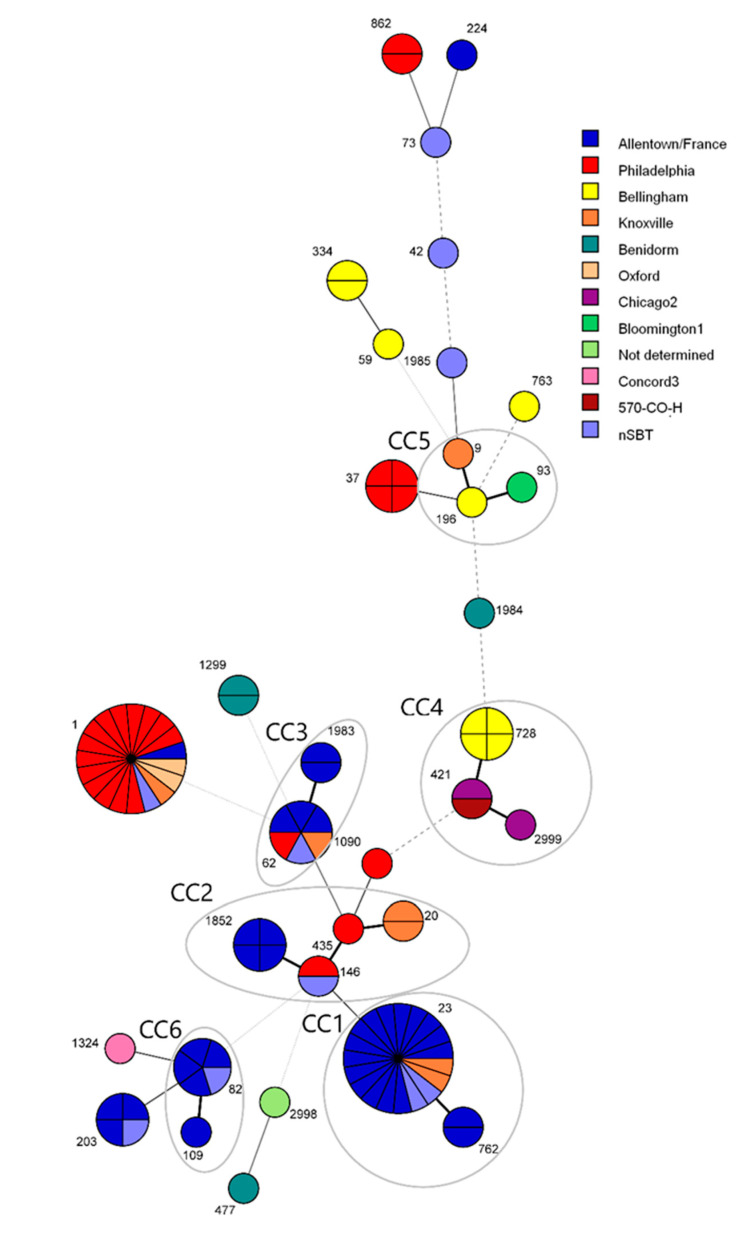
Minimum spanning tree of 88 *Legionella pneumophila* clinical isolates, together with 10 profiles obtained by nested SBT (nSBT). The sizes of circles indicate ST sample sizes, while lengths of lines connecting STs indicates extent of variation. STs connected by solid lines indicate one, two, or three allele changes (the shortest thick black line denotes single-locus variants), grey dashed lines represent four changes, and grey dotted lines represent five or more changes. The clonal complexes (CCs) of single-locus variants are circled.

**Table 1 diagnostics-11-01201-t001:** Presentation of 88 *Legionella pneumophila* isolates obtained from clinical specimens with listings of phenotypes and sequence types (STs). *L. pneumophila* sg, *Legionella pneumophila* serogroup; *n*, number; ND, not determined.

*L. pneumophila* Serogroup (*n*/%)	Phenotype (*n*/%)	Sequence Types (*n*)
*L. pneumophila* sg 1 (82/93.2%)	Allentown/France (36/43.9%)	ST23 (15); ST82 (4); ST1852 (4); ST62 (3); ST203 (3); ST762 (2); ST1983 (2); ST1 (1); ST109 (1); ST224 (1);
	Philadelphia (24/29.3%)	ST1 (14); ST37 (4); ST862 (2); ST62 (1), ST146 (1); ST435 (1); ST1090 (1)
	Bellingham (9/11.0%)	ST728 (4); ST334 (2); ST59 (1); ST196 (1); ST763 (1)
	Knoxville (7/8.5%)	ST1 (1); ST9 (1); ST20 (2); ST23 (2); ST62 (1)
	Benidorm (4/4.9%)	ST1299 (2); ST477 (1); ST1984 (1)
	Oxford/Olda (2/2.4%)	ST1 (2)
*L. pneumophila* sg 2-14 (6/6.8%)	Bloomington 1 (1/16.7%)	ST93 (1)
	Chicago 2 (2/33.3%)	ST421 (1); ST2999 (1)
	Concord 3 (1/16.7%)	ST1324 (1)
	570 CO H (1/16.7%)	ST421 (1)
	ND (1/16.7%)	ST2998 (1)

## Data Availability

The datasets used and analyzed during the current study are available from the corresponding author on reasonable request.

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
