# Peer review of "Legionella pneumophila—Epidemiology and Characterization of Clinical Isolates, Slovenia, 2006–2020"

_diagnostics, 2021, doi:10.3390/diagnostics11071201_

Round 1

Reviewer 1 Report

Manuscript entitled “Legionella pneumophila – epidemiology and characterization of clinical isolates, Slovenia., 2006-2020, by Kese et al reports data of Legionella pneumophila diagnosis and typing in patients hospitalized with pneumonia symptoms in Slovenia and mainly in Ljubljana.

Although the manuscript could be written better, it is quite satisfying for materials and results.

It should be interesting to have information about the origin of these infections, that is if they were hospital, travel or community-acquired, if they were part of outbreak, cluster or they were sporadic cases, and if there were cases with an ascertained source of infection.

More specific comments

As the authors state at the end of the manuscript, a limit of the study was do not have a complete representativeness of the country. For this reason, it would be useful a map of the described cases, in order to understand also a possible gradient of LD notification.

In 2.3 PCR section line 85, please explain better “r-gene Legionella pneumophila….”, this does not appear specific to Legionella.

In section 2.4, the authors describe culture examination. Did they follow an ISO norm or other, was culture based on a standardized method? In addition, how many patients were diagnosed by culture? Was it performed only retrospectively on respiratory samples stored at -80°C. Please, specify this. Is this the reason for which culture is not in Figure 1? Authors should explain better.

Most cases were MAb3/1 positive and they were prevalently associated to ST1 and ST23: also here it would be interesting to have information with the origin of infection.

The authors should avoid to repeat results in discussion while they could compare similarity and differences with other European countries, especially concerning the STs circulating.

It is recommended an English revision.

Author Response

Response to Reviewer 1 Comments

Thank you for your helpful comments and suggestions, which we have carefully addressed in our reply. Please find below our specific responses to your questions and comments.

Point 1: Manuscript entitled “Legionella pneumophila – epidemiology and characterization of clinical isolates, Slovenia, 2006-2020, by Kese et al reports data of Legionella pneumophila diagnosis and typing in patients hospitalized with pneumonia symptoms in Slovenia and mainly in Ljubljana.

Although the manuscript could be written better, it is quite satisfying for materials and results. It should be interesting to have information about the origin of these infections, that is if they were hospital, travel or community-acquired, if they were part of outbreak, cluster or they were sporadic cases, and if there were cases with an ascertained source of infection.

Response 1:  We thank the reviewer for pointing out the very important issue. In this retrospective study we wished to present data more from the microbiological point of view and so we have not gathered epidemiological and clinical data for all the included patients which is certainly one of the limitations of our study. In the chapter 4. Discusion (Lines 1117-1118), we added information that Legionella cases were mostly sporadic and community acquired. We also explained two minor outbreaks we observed in designated period of time (Lines 1128-1135).

More specific comments

Point 2: As the authors state at the end of the manuscript, a limit of the study was do not have a complete representativeness of the country. For this reason, it would be useful a map of the described cases, in order to understand also a possible gradient of LD notification.

Response 2: Thank you for this suggestion. In chapter 2.1 Sample collection and routine Legionella testing, we have added information from which hospitals and the Statistical Regions treated patients (lines 246-250). We also provide information on how many residents live in Statistical Regions compared to the total population.

Point 3: In 2.3 PCR section line 85, please explain better “r-gene Legionella pneumophila….”, this does not appear specific to Legionella.

 Response 3: Thank you. This has been amended and the sentence now reads as follows:

Real-time PCR targeting the mip gene of L. pneumophila (Argene-BioMerieux diagnostics, France) was performed on a LightCycler 2.0 (Roche, Germany) platform according to the manufacturer´s instructions.

Point 4: In section 2.4, the authors describe culture examination. Did they follow an ISO norm or other, was culture based on a standardized method? In addition, how many patients were diagnosed by culture? Was it performed only retrospectively on respiratory samples stored at -80°C. Please, specify this. Is this the reason for which culture is not in Figure 1? Authors should explain better.

 Response 4: Thank you. As suggested by the reviewer, we now provide a description about the Legionella spp. cultivation, which reads as follows:

Legionella cultivation was subsequently performed only on the remaining respiratory samples from PCR-positive patients. Sputum samples were liquefied with Sputasol (OXOID, U.K.) in a 1:1 ratio. After centrifugation at 1000 xg for 15 min, the supernatant was discarded. The resulting pellet was resuspended in 1mL of sterile H2O and heat-treated at 50 °C for 30 minutes. Buffered charcoal–yeast extract agar (BCYE) and Legionella selective (BMPA) agar plates with antibiotics (OXOID, U.K.) were inoculated using the standard technique [14]. Plates were incubated aerobically in a humid atmosphere at 35–37 °C for up to 10 days with regular macroscopic and microscopic inspections for specific Legionella colonies every two days. Presumptive Legionella colonies that appeared after the third day with a round, smooth shape and a bluish-gray, or whitish appearance were identified using a Microflex LT (Bruker Daltonics, USA) MALDI-TOF MS system according to the manufacturer´s instructions. Data acquisition and result interpretation were automatically performed by the MALDI Biotyper software (Bruker Daltonics, USA). All Legionella isolates were identified by indirect immunofluorescence test (IFA) using specific monoclonal antibodies and then stored on –80 °C.

Point 5: Most cases were MAb3/1 positive and they were prevalently associated to ST1 and ST23: also here it would be interesting to have information with the origin of infection.

 Response 5: Yes, we fully agree with you, but since the study is retrospective, we unfortunately have no information about the origin of the infection. We only know that the cases were mostly sporadic and CAP.

Point 6: The authors should avoid to repeat results in discussion while they could compare similarity and differences with other European countries, especially concerning the STs circulating.

 Response 6: We thank you. This has been taken into account.

Point 7: It is recommended an English revision.

Response 7: We thank you. This has been considered. The English review was conducted by MDPI English Editing Service.

All the changes in the manuscript are recorded by the “Track Changes” function.

Best Regards,

Darja Keše on behalf of all authors

Reviewer 2 Report

Dear authors

The topic of Legionellae is of increasing importance in Europe, as more and more cases occur. Your epidemiologic study of diagnostic outcomes in hospital environment over the last 15 years including characterization of strains provides a good overview of the legionella situation and its development in your country. It is presented clearly and simply. To further improve your work, I recommend more detailed description of Methods used, to provide better overview to the reader. Furthermore, I recommend adding suggestions to improve clinical diagnostics in future.

Minor revisions

  • Improve the Material and Method sector by including more details about the analyzes used in order to give a better insight to the reader and to spare them excessive use of references and google. So readers will need the references only for the very details.

And add information, if used methods are validated or not, above all for PCR detection and culture.

  • In addition, I recommend providing some recommendations for future LD diagnostics.

Minor revisions

L24 include "epidemiology" and optionally "pneumophila" in keywords

L27 parasites (plural)

L38/39 use CAP as abbreviation.

L41-44 Reference this statement

L45 what about other Balkan countries, UK and Switzerland, that are no (longer) EU members? Country?

L54-57 Divide this long sentence in more than one

L65 delete point. Also in further following headings

L66 add hospitalized ( …from 15540 hospitalized patients…)

L68 give range: 18 – xx years of age (median age was 62)

L79 specify what Legionella species and serogroups of L. pneumophila these test detect

L86 indicate company and what gene the ARGENE assay detects. It might be the mip-gene. R-gene is not a scientific gene naming

L87 although reference is given, specify what type of analysis (qPCR) this additional assay was.

L91 add Incubation and assessment criteria

L106 Use italic font for Legionella consistently, throughout the manuscript

L126 add more information to the title, above all the Legionella subject to render the figure self-explanatory, and use same abbreviations as in text (e.g. UAT), but still describe them. Figures have to be independent from text body.

L122 seroconversion (add i)

L123 replace patients with sera (or serum samples)

L124 516/717 = 72.0%

L125 What about the ratio of CAP / HAP ?

L129 Add Legionella subject (see comment on L126)

L135 percentage = positivity rate?

L140 see L219

L142-155 these to figures could be merged, as the blue bars are twice the same. Lines could be presented in different colors.

L165 as the additional PCR only detects sg1, I would prefer you write: … and 11 (8.5%) were negative for sg 1.

L166-168 to me it is not clear, if the sg percentages are based on PCR detection of sg1 (negative one's are 2-14), or on IFA, DFA, or on Dresden panel

L174 9/82 = 11.0% (same in table1)

L177 Table 1's headline: replace "No" with "n". And 1st row: 82/88 = 93.2%

L182 the ST728 does not appear in Figure5. And it contributes 4.1% not 4.4%.

L195 Figure 5: I guess the 428 should be the 728 (see comment on L182)

L183/184 describe the digits in the brackets

L189 uses CC instead of clonal group

L204 why are the 10 additional STs (obtained by nested SBT) not included in the tree?

L207 Figure 6: Add all 98 ST this Figure. Indicate difference of full and dotted lines. And please do not cover ST numbers by CC circles.

If only the 88 STs are included in this Figure, why are there Subtypes that are missing in Table 1 (e.g. Chicago, Concord, Bloomington)?

L217 Legionella pneumophila isolates

L231 Add reference no. for Dagan et al.

L233-235 Give more emphasis to this statement, as it is crucial to further evaluate patients with clinical sign of LD, when UAT is negative.

L240 what about the ratio of CAP / HAP?

L247 climate change (singular)

L266 No comma in 100 000 (or equally throughout the manuscript)

L269 emphasize the high percentage of sg1 (Almost 93.2 % of clinical isolates belonged to sg1 of L. pneumophila…) as of course you found only pneumophila

L273 Mention that ST1 (and ST23) do belong to different phenotypes.

L278 Delete space in ST37

L283-285 for research reason, a PCR for Legionella spp could be performed, as DNA is anyway extracted from respiratory samples for the ARGENE PCR. E.g. Nazarian et al. (2008) published such a PCR for L. spp detecting the 23S rRNA gene.

Discuss additionally, that the STs are not matching to the phenotypes obtained by the Dresden panel.

Best regards

Author Response

Response to Reviewer 2 Comments

We thank Reviewer #2 for their positive comments and a detailed review of the manuscript. We are grateful for the issues raised as they helped to improve our manuscript and clarify some potential ambiguities. Please find below our specific responses to your questions and comments.

The topic of Legionellae is of increasing importance in Europe, as more and more cases occur. Your epidemiologic study of diagnostic outcomes in hospital environment over the last 15 years including characterization of strains provides a good overview of the legionella situation and its development in your country. It is presented clearly and simply. To further improve your work, I recommend more detailed description of Methods used, to provide better overview to the reader. Furthermore, I recommend adding suggestions to improve clinical diagnostics in future.

Response: We thank you for your helpful comments. At the reviewer's suggestion, we have described the methods used in more detail in the "Materials and Methods" section, particularly PCR and culturing. At the end of the "Discussion" section, we have also added some future suggestions to improve our work.

Minor revisions

Point 1: Improve the Material and Method sector by including more details about the analyzes used in order to give a better insight to the reader and to spare them excessive use of references and google. So readers will need the references only for the very details.

Response 1: Thank you. In chapter 2. Materials and Methods we describe the methods used in more detail according to your suggestions.

Point 2: And add information, if used methods are validated or not, above all for PCR detection and culture.

Response 2: Thank you. The information has been added according to your suggestions.

Real-time PCR targeting the mip gene of L. pneumophila (Argene-BioMerieux diagnostics, France) was performed on a LightCycler 2.0 (Roche, Germany) platform according to the manufacturer´s instructions. PCR-positive samples were further analyzed by interlaboratory-validated L. pneumophila triplex qPCR assay to discriminate between L. pneumophila sg1 (wzm target gene) and L. pneumophila (mip gene) on the Rotor-Gene Q (Qiagen, Germany) platform as previously described [13]. In both PCR tests internal controls were included.

Legionella cultivation was subsequently performed only on the remaining respiratory samples from PCR-positive patients. Sputum samples were liquefied with Sputasol (OXOID, U.K.) in a 1:1 ratio. After centrifugation at 1000 xg for 15 min, the supernatant was discarded. The resulting pellet was resuspended in 1mL of sterile H2O and heat-treated at 50 °C for 30 minutes. Buffered charcoal–yeast extract agar (BCYE) and Legionella selective (BMPA) agar plates with antibiotics (OXOID, U.K.) were inoculated using the standard technique [14]. Plates were incubated aerobically in a humid atmosphere at 35–37 °C for up to 10 days with regular macroscopic and microscopic inspections for specific Legionella colonies every two days. Presumptive Legionella colonies that appeared after the third day with a round, smooth shape and a bluish-gray, or whitish appearance were identified using a Microflex LT (Bruker Daltonics, USA) MALDI-TOF MS system according to the manufacturer´s instructions.

Point 3: In addition, I recommend providing some recommendations for future LD diagnostics.

Response 3: Thank you. We added some suggestions for future LD diagnostics at the end of Discusion.

Minor revisions

Point 4: L24 include "epidemiology" and optionally "pneumophila" in keywords

Response 4: Thank you for your suggestion. Done as requested.

Point 5: L27 parasites (plural)

Response 5: Thank you! Corrected as suggested.

Point 6: L38/39 use CAP as abbreviation.

Response 6: Thank you! CAP is used as abbreviation.

Point 7: L41-44 Reference this statement

Response 7: Thank you! Reference 7 was added.

Point 8: L45 what about other Balkan countries, UK and Switzerland, that are no (longer) EU members? Country?

Response 8: Thank you! We also added the UK and Switzerland and reference. For the other Balkan countries we did not get any references.

Fischer, F.B.; Schmutz, C.; Gaia, V.; Mäusezahl, D. Legionnaires’ Disease on the Rise in Switzerland: A Denominator-Based Analysis of National Diagnostic Data, 2007–2016. International Journal of Environmental Research and Public Health 2020, 17, 7343, doi:10.3390/ijerph17197343.

Point 9: L54-57 Divide this long sentence in more than one

Response 9: Thank you! Done as suggested.

Point 10: L65 delete point. Also in further following headings

Response 10: Thanks you. Point was deleted in all headings.

Point 11: L66 add hospitalized ( …from 15540 hospitalized patients…)

Response 11: All patients were not hospitalized, that is why we wroteDepending on the severity of symptoms, patients were treated as inpatients or outpatients.«

Point 12: L68 give range: 18 – xx years of age (median age was 62)

Response 12: Thank you. Range was added.

Point 13: L79 specify what Legionella species and serogroups of L. pneumophila these test detect

Response 13: Requested specification was added.

Point 14: L86 indicate company and what gene the ARGENE assay detects. It might be the mip-gene. R-gene is not a scientific gene naming

Response 14: Information was added.

Point 15: L87 although reference is given, specify what type of analysis (qPCR) this additional assay was.

Response 15: Thank you. Information was added.

Point 16: L91 add Incubation and assessment criteria

Response 16: Thank you. The cultivation method was described in more detail and the required information was added.

Point 17: L106 Use italic font for Legionella consistently, throughout the manuscript

Response 17: Thank you for your comment. Italic font was used throughout the manuscript.

Point 18: L126 add more information to the title, above all the Legionella subject to render the figure self-explanatory, and use same abbreviations as in text (e.g. UAT), but still describe them. Figures have to be independent from text body.

Response 18: Thank you! Done as suggested.

Point 19: L122 seroconversion (add i)

Response 19: 'i' was added

Point 20: L123 replace patients with sera (or serum samples)

Response 20: Thank you! It was replaces as suggested.

Point 21: L124 516/717 = 72.0%

Response 21: Thank you! It was corrected.

Point 22: L125 What about the ratio of CAP / HAP ?

Response 22: Thank you very much for the question. You are right. It would certainly be interesting to determine the relationship between CAP and HAP, but unfortunately we do not have information on the origin of LD for each patient.

Point 23: L129 Add Legionella subject (see comment on L126)

Response 23: Done as suggested.

Point 24: L135 percentage = positivity rate?

Response 24: Thank you! It was corrected.

Point 25: L140 see L219

Response 25: Done as suggested.

Point 26: L142-155 these to figures could be merged, as the blue bars are twice the same. Lines could be presented in different colors.

Response 26: Thank you for your suggestion. Both figures were merged. Please see Figure 3.

Point 27: L165 as the additional PCR only detects sg1, I would prefer you write: … and 11 (8.5%) were negative for sg 1.

Response 27: In this revision, we described a triplex qPCR test that differentiates between L. pneumophila sg 1 (target gene wzm) and L. pneumophila (mip gene). For this reason, we did not change the sentence.

Point 28: L166-168 to me it is not clear, if the sg percentages are based on PCR detection of sg1 (negative one's are 2-14), or on IFA, DFA, or on Dresden panel

Response 28: Thank you. To make it clearer, we added an absolute number.

Point 29: L174 9/82 = 11.0% (same in table1)

Response 29: Thank you for correction. Corrected accordingly in text and in Table 1.

Point 30: L177 Table 1's headline: replace "No" with "n". And 1st row: 82/88 = 93.2%

Response 30: Thank you for correction. Done as suggested.

Point 31: L182 the ST728 does not appear in Figure5. And it contributes 4.1% not 4.4%.

Response 31: Thank you for correction. Figure 5 was corrected. Percentages in the text were corrected.

Point 32: L195 Figure 5: I guess the 428 should be the 728 (see comment on L182)

Response 32: Thank you for noticing. ST number was corrected.

Point 33: L183/184 describe the digits in the brackets

Response 33: The digits were allel numbers, we decided to erase them.

Point 34: L189 uses CC instead of clonal group

Response 34: Thank you for correction. Done as suggested.

Point 35: L204 why are the 10 additional STs (obtained by nested SBT) not included in the tree?

Response 35: Thank you for your suggestion. We have added 10 additional STs in the MST.

Point 36: L207 Figure 6: Add all 98 ST this Figure. Indicate difference of full and dotted lines. And please do not cover ST numbers by CC circles.

Response 36: Thank you for suggestion. Done as suggested. Please see Figure 6.

Point 37: If only the 88 STs are included in this Figure, why are there Subtypes that are missing in Table 1 (e.g. Chicago, Concord, Bloomington)?

Response 37: Thank you for suggestion. Table 1 was updated.

Point 38: L217 Legionella pneumophila isolates

Response 38: Thank you. It was corrected.

Point 39: L231 Add reference no. for Dagan et al.

Response 39: Thank you. Reference was added.

Point 40: L233-235 Give more emphasis to this statement, as it is crucial to further evaluate patients with clinical sign of LD, when UAT is negative.

Response 40: Done as suggested.

Point 41: L240 what about the ratio of CAP / HAP?

Response 41: In the study of Muyldermans, A. they do not look for the ratio CAP/HAP.

Point 42: L247 climate change (singular)

Response 42: Thank you. It was corrected.

Point 43: L266 No comma in 100 000 (or equally throughout the manuscript)

Response 43: Corrected.

Point 44: L269 emphasize the high percentage of sg1 (Almost 93.2 % of clinical isolates belonged to sg1 of L. pneumophila…) as of course you found only pneumophila

Response 44: This part was rewritten.

Point 45: L273 Mention that ST1 (and ST23) do belong to different phenotypes.

Response 45: Thank you. Done as suggested.

Point 46: L278 Delete space in ST37

Response 46: Done as suggested.

Point 47: L283-285 for research reason, a PCR for Legionella spp could be performed, as DNA is anyway extracted from respiratory samples for the ARGENE PCR. E.g. Nazarian et al. (2008) published such a PCR for L. spp detecting the 23S rRNA gene.

Response 47: Thank you for suggestion. We will try to do this.

Point 48: Discuss additionally, that the STs are not matching to the phenotypes obtained by the Dresden panel.

Response 48: Thank you. Done as suggested.

All the changes in the manuscript are recorded by the “Track Changes” function.

Best Regards,

Darja Keše on behalf of all authors